# The Influence of Curing System on the Macroscopic Performance and Microstructure of Anti-Abrasive UHPC

**Jinhui Li [1], Zi Yu [1], Fang Xu [2],\*, Zhijiong Guo [2] and Qingjun Ding [3]**

[1] College of Materials Science and Engineering, Wuhan Textile University, Wuhan 430200, China; jhli@wtu.edu.cn (J.L.); yu1018541283@163.com (Z.Y.)
[2] Faculty of Engineering, China University of Geosciences, Wuhan 430074, China; 20220901gzj@cug.edu.cn
[3] School of Materials Science and Engineering, Wuhan University of Technology, Wuhan 430070, China; dingqj2003@whut.edu.cn
\* Correspondence: xufang@cug.edu.cn

**Abstract:** In a previous study, we utilized saturated prewet high titanium heavy slag sand to produce UHPC (ST-UHPC). ST-UHPC has high impact and abrasion resistance. For better ST-UHPC applications, we investigate the mechanism of ST-UHPC under different curing systems from the microstructure and macroscopic perspective in this paper. We prepared ST-UHPC under four maintenance conditions: 20 °C standard curing, 90 °C steam curing, 90 °C dry curing and 210 °C 2 MPa pressure steam curing. Then, we analyzed the hydration product composition, the degree of cement hydration, the C-A-S-H gel microstructure and the substitution of $Al^{3+}$ for $Si^{4+}$ in relation to these prepared ST-UHPCs. Compared with standard curing, dry curing at 90 °C accelerated the water evaporation and reduced the hydration degree of ST-UHPC cementite. However, pressure steam curing significantly improved the hydration degree of ST-UHPC cementing material, and increased the MCL and Al[4]/Si of C-A-S-H gel. In addition, pressure steam curing reduced the Ca/Si and promoted the conversion of C-A-S-H cementing to tobermorite. Compared with dry curing at 90 °C, pressure steam curing significantly improved the macroscopic properties of ST-UHPC. The macro-performance difference of ST-UHPC under standard curing and 90 °C steam curing is small. The reason is that steam curing caused the water to be rapidly released in the internal aggregate of ST-UHPC. This resulted in the increase of the interface between the internal aggregate of ST-UHPC and the ST-UHPC cementate. The harmful pores in the ST-UHPC matrix under steam curing were also increased. To sum up, compared with standard curing, dry curing at 90 °C weakened the mechanical properties and microstructure of ST-UHPC, but steam pressure curing increased them. The single steam curing had no significant effect on the mechanical properties and microstructure of the ST-UHPC. Therefore, non-steam and room-temperature moisturizing maintenance should be adopted for anti-abrasive UHPC.

**Keywords:** curing system; porous high titanium heavy slag sand; anti-impact-wear UHPC; microstructure; macro-performance; influence mechanism

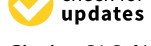


## 1. Introduction

The mountainous regions in the western part of the country are areas of more difficult terrain, which are subject to rainfall and snowmelt all year round, and the mountains are prone to the phenomenon of mud and rock scouring. This natural process has a series of effects on the geological environment, one of which is damage to infrastructure such as buildings and bridges, especially the wear and tear of concrete columns.

Rain and snowmelt trigger mud and rock scouring: the terrain of the western mountains is highly undulating, and rain and snowmelt carry away large quantities of soil and rocks as they flow between the mountains, forming mud and rock flows. This high-speed flow of mud and rocks has a strong destructive force and can easily wash away the protective layer of the surface of the concrete column, exposing the internal column.

This leads to concrete surface abrasion: during the process of scouring of debris, the sand, gravel particles and soil carried by the debris produce a strong impact and abrasive effect on the concrete surface, resulting in the abrasion of the concrete surface. This abrasion not only makes the concrete surface become uneven, but may also expose the internal aggregate of the concrete, weakening the overall strength of the concrete.

Structural damage is caused to concrete columns: prolonged debris scouring causes the surface of concrete columns to gradually lose its protection, and the deeper structure is damaged. Mudslides can cause cracks in concrete columns, which in turn accelerate the deterioration and embrittlement of the concrete, and may eventually lead to the collapse of the columns [1–4].

Ultra-High Performance Concrete (UHPC), also known as Reactive Powder Concrete (RPC), is the most innovative cementitious engineering material of the past three decades [5–8], compatible with the first patent filed by Hans Henrik Bache of Denmark in 1979 and his establishment of the DSP theory [9–12]. The addition of different aggregates and admixtures will affect the performance of UHPC [13–16]. Impact- and abrasion-resistant UHPC provides new ideas for bridge piers and columns in western mountainous areas. UHPC is based on the tightest stacking theory [17–19]. In recent years, there has been great progress in research into the material design, preparation, properties and engineering application of UHPC [20]. But UHPC is rarely used in the anti-abrasive protection engineering of bridges in western mountainous areas. The problems of technology preparation of high-impact-wear UHPC have not been solved. In addition, the microstructure evolution and property degradation mechanism of UHPC under impact wear and erosion have not been researched. Furthermore, the microstructure control mechanism of high-impact-wear UHPC material and the engineering application technology still need to be studied. The above problem seriously restricts the application of high-impact-wear UHPC in the anti-abrasive protection engineering of bridge piers and columns in western mountainous areas.

Aiming to solve the durability problem of a bridge pier damaged by abrasion in a western mountainous area, we put forward ST-UHPC. The preparation of the ST-UHPC was based on the optimization design of material composition and microstructure control. The ST-UHPC was developed by using saturated prewet high-strength wear-resistant porous high-titanium heavy slag sand and modified rubber particles. The modified rubber particles are mainly used to improve the properties of UHPC by first changing the surface morphology of rubber particles, and then improving the surface hydrophilicity and the interface bonding properties of the rubber particle-cementing material matrix to strengthen the UHPC [21]. The internal curing and arch effect of prewet high titanium heavy slag sand, as well as the superposition effect of internal curing and expander of prewet high titanium heavy slag sand, further improved the anti-abrasion and volume stability of the UHPC [12]. The prepared anti-abrasive UHPC has high strength, low shrinkage, high abrasion resistance, impact toughness and high durability [22]. But ST-UHPC needs to be maintained after pouring when applied to practical engineering. The research shows that there are great differences in the microstructure and macroscopic properties of UHPC under different curing systems. The performance improvement of ST-UHPC directly depends on the curing system. Therefore, there is an urgent need to study the influence of the curing system on the microstructure and properties of ST-UHPC, in order to guide the maintenance construction of UHPC protective materials in engineering.

A UHPC high-temperature and high-pressure curing system usually includes high-temperature steam curing and pressure steam curing [23]. The temperature of high-temperature steam curing is usually 60~90 °C, the time is 48~72 h and the curing cycle is long [24]. However, the pressure steam curing temperature is generally 180~230 °C, the pressure is 1~2 MPa and the curing time is 4~16 h. Pressure steam curing shortened the curing period greatly. The mechanical properties of UHPC obtained under pressure steam curing are better than those obtained under steam curing [25].

He Feng [26] studied the variation rules of the mechanical properties of UHPC under curing systems such as 90 °C hot water curing for 48 h followed by 20 °C water standard curing, 90 °C hot water curing for 48 h followed by 200 °C dry heat curing for 48 h, etc. He found that the compressive strength of concrete under high temperature curing was higher than under hot water curing, which was higher than under standard curing. But different curing systems had little influence on the flexural strength of UHPC. When the water–binder ratio was 0.35, He Feng prepared UHPC with a compressive strength of 135 MPa. Talebinejad I et al. [27] studied the influence of curing system and water–binder ratio on the macroscopic mechanical properties of UHPC. When cement and silica ash were used as cementation materials and the cement dosage was 1900 $kg/m^3$, the strength of the UHPC was up to 230 MPa under the condition of 20 °C followed by 90 °C. When the water–binder ratio was reduced to 0.11 and the steam curing system was adopted at 200 °C, the compressive strength of the prepared UHPC reached 330 MPa. Yazici H, Deniz E et al. [28] studied the influence of pressure, temperature and time of steam curing on the mechanical properties of UHPC. Yazici H believed that pressure, temperature and time of steam curing had an important influence on the properties of UHPC. And for a certain pressure and temperature, Yazici H thought there was an optimal value of steam curing time for UHPC. Muller U et al. [29] studied the influence of pressure steam curing time on the mechanical properties of UHPC. They found that the strength of the UHPC reached its maximum value when the curing time was 10–24 h. They believed that the strength of the UHPC even decreased slightly when the pressure steaming time increased. Muller U analyzed the microstructure at the overlong pressure steaming time. It was found that the hydration products inside the UHPC crystallize excessively and the size increases. This results in micro-cracks at the interface. Therefore, this is not conducive to the development of UHPC strength. Shi C, Mindess S et al. [30,31] believed that for UHPC with different temperatures and pressure curing systems, there is an optimal curing period when its mechanical properties reach the optimal value.

C-S-H gel is the main source of concrete strength. Under different curing systems, the microstructure of C-S-H gel will change significantly. Thus, different curing systems affecting the macro-properties of UHPC [32]. At present, there are few studies on the microstructure of UHPC under different curing systems. An Mingzhe et al. [33] studied the effect of high-temperature steam curing time on the microstructure of UHPC. The results showed that C-S-H gels presented different particle morphology under high-temperature steam curing of 1d, 2d and 3d. The morphologies of the C-S-H gels, respectively, were small particle, worm-like and cloud-like. The morphology gradually develops from loose shape to dense mass. Therefore, An Mingzhe confirmed that the extension of high-temperature steam curing time could improve the concrete's compactness. Hong S Y, Glasser F P et al. [34] studied the phase transition of $CaO\text{-}SiO_2\text{-}H_2O$ system at 85 °C~200 °C by the synthesis method. He found that the hydration products of the $CaO\text{-}SiO_2\text{-}H_2O$ system were formed into aquamarite, afwillite, xonotlite, tobermorite and jennite, respectively, under different temperatures. He pointed out that tolbermorite would decompose into xonotlite above 130 °C. The stable temperature range for xonotlite existing was 85 °C~200 °C. Jennite would decompose into afwillite and xonotlite at 150 °C. Jennite would also decompose into afwillite and anthracite above 160 °C. The stable temperature range for afwillite existing was 160 ± 10 °C. But afwillite would decompose into aquamarite and anthracite at high temperatures. Aquamarite was stable in a very small temperature range, with an upper temperature of 210 °C. Above 210 °C, aquamarite will decompose into $\alpha$-$C_2SH$ ($Ca_2(HSiO_4)(OH)$). Halit Yazici [35] studied the mechanical properties and microstructure evolution of UHPC at 210 °C and 2 MPa. Halit Yazici found that C-S-H gel transformed into $\alpha$-$C_2SH$ under pressure steaming conditions. A-$C_2SH$ improved the densification of UHPC, but slightly reduced the mechanical properties of UHPC. In the presence of mineral admixtures such as silica fume and fly ash, $\alpha$-$C_2SH$ transformed into tobermorite. At the same time, with the extension of pressure evaporation time, $\alpha$-$C_2SH$ transformed into other crystalline hydration products. Reduction of $\alpha$-$C_2SH$ led to a decrease in the mechanical

properties of UHPC. When C-S-H gel transformed into tobermorite completely, the mechanical properties of UHPC were weakened. When the crystallized product generated reaches a certain extent, it may adversely affect the mechanical properties of UHPC [36–38]. At the same time, when C-S-H gel converted into tobermorite completely, it had a negative impact on the durability of UHPC.

In summary, steam curing improved the hydration degree of UHPC. Compared with standard curing, the mechanical properties and durability of UHPC were improved significantly under 90 °C steam curing. Steam curing improved the hydration degree of UHPC cementing material significantly, and promoted the conversion of C-A-S-H cementing to tobermorite. Steam curing could improve the mechanical properties and durability of UHPC. Dry heat curing reduced the hydration degree of UHPC and the mechanical properties and durability of UHPC. However, the above-mentioned effects of the curing system on the properties of UHPC mainly focus on the macroscopic mechanical perspective. The effects of the curing system on the properties and microstructure of UHPC has not been analyzed thoroughly from the angle of the microscopic molecules.

Most of the above studies focused on common aggregates. There was no study on the influence mechanism of the curing system of ST-UHPC by porous curing. Under high-temperature and steam curing, the moisture will evaporate inside the saturated prewet high-strength wear-resistant porous aggregate (SP). The influence law of the SP on the performance of UHPC needs to be further studied. Whether the influence law of the SP on the performance of UHPC is different from the ordinary aggregate also needs to be further studied. This is also the focus of this paper.

In this paper, in order to investigate the mechanism of ST-UHPC under different curing systems from the microstructure and macroscopic perspectives, we prepared UHPC under four maintenance conditions: 20 °C standard curing, 90 °C steam curing, 90 °C dry curing and 210 °C 2 MPa pressure steam curing. Then we analyzed the hydration product composition, the degree of cement hydration, the C-A-S-H gel microstructure and the substitution of $Al^{3+}$ for $Si^{4+}$ in relation to these prepared UHPCs. In addition, the effects of the curing system on the mechanical properties, impact-wear properties and durability of UHPC were also studied.

## 2. Experimental Program

### 2.1. Materials

#### 2.1.1. Cementitious Material

In this paper, cement, fly ash microbeads and silica fume are applied as cementitious materials. The appearance of the cementitious materials is shown in Figure 1.

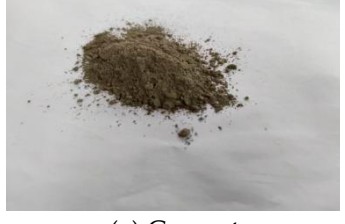 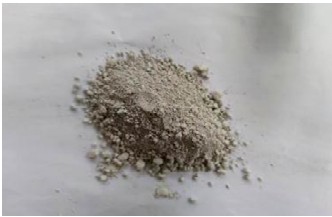 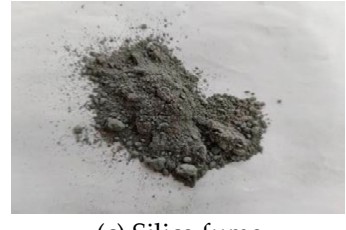

(**a**) Cement        (**b**) Fly ash microbeads        (**c**) Silica fume

**Figure 1.** The appearance of the cementitious materials.

(1) Cement: A high iron phase cement with high corrosion resistance and wear resistance was produced industrially by Guangxi Yufeng Cement Co., Ltd. (Liuzhou, China). The cement was refined twice (HFC1).

(2) Fly ash microbeads (FAM): Fly ash microbeads were produced by Tianjin Zhuancheng New Materials Co., Ltd. (Tianjing, China). The specific surface area of FAM was greater than or equal to 1200 $m^2$/kg. The 28d activity index of FAM was greater than or equal to 90%.

(3)   Silica fume (SF): The silica fume was produced by Shanghai Tiankai Company (Shanghai, China). The specific surface area of SF was 19,500 $m^2$/kg. The $SiO_2$ content of SF was 96.3%.

2.1.2. Chemical Analysis of Cementitious Materials

The chemical analysis of HFCL, FAM and SF is shown in Table 1.

**Table 1.** Chemical analysis of HFCL, FAM and SF.

| Type | SiO$_2$ | Al$_2$O$_3$ | Fe$_2$O$_3$ | MgO | CaO | Na$_2$O | K$_2$O | SO$_3$ | P$_2$O$_5$ | Loss |
|------|---------|-------------|-------------|-----|-----|---------|--------|--------|------------|------|
| HFC1 | 21.61 | 3.14 | 6.17 | 1.98 | 62.32 | 0.16 | 0.37 | 2.16 | 0.14 | 1.72 |
| FAM | 34.41 | 18.19 | 4.38 | 5.53 | 1.38 | 1.49 | 0.94 | / | / | 2.4 |
| SF | 96.34 | 0.61 | 0.16 | 0.25 | 0.54 | 0.08 | 0.21 | 0.13 | / | 1.68 |

2.1.3. Physical Properties of the Three Cementitious Materials

The physical properties of the three cementitious materials are analyzed in Table 2. Add 10% cement mass of FAM and SF into the cement, and then stir the sample for relevant performance testing.

**Table 2.** Physical properties of the three cementitious materials.

| Materials | Specific Surface Area/(cm$^2$/g) | Initial Setting Time/min | 3d Compressive Strength/MPa | 28d Compressive Strength/MPa |
|-----------|----------------------------------|--------------------------|------------------------------|-------------------------------|
| HFC1 | 3580 | 195 | 25.8 | 59.3 |
| FAM | 12,000 | 130 | 30 | 51.4 |
| SF | 19,500 | 255 | 34.5 | 86.1 |

Note: The corresponding dosage of initial setting time, 3d compressive strength and 28d compressive strength to FAM and SF are both 10%.

2.1.4. High Titanium Heavy Slag Sand (HTHSS)

The high titanium heavy slag sand was produced by Anning Titanium Technology Co., Ltd. in Panzhihua, Sichuan. The apparent density of HTHSS was 3100 kg/m$^3$. The fineness modulus of HTHSS was 3.0. Table 3 shows the results of X-ray fluorescence physical properties. Table 4 shows the physical properties of HTHSS.

**Table 3.** Main chemical compositions of high titanium heavy slag sand/%.

| Raw Material | SiO$_2$ | Al$_2$O$_3$ | Fe$_2$O$_3$ | CaO | MgO | TiO$_2$ | FeO | mFe | Loss |
|--------------|---------|-------------|-------------|-----|-----|---------|-----|-----|------|
| High titanium heavy slag sand | 18.09 | 14.85 | 5.31 | 24.65 | 8.50 | 22.65 | 2.55 | 2.12 | 1.28 |

**Table 4.** Physical property parameters of high titanium heavy slag sand.

| Test Item | Apparent Density/kg/m$^3$ | Stone Powder Content/% | Water Absorption Rate/% | Crushing Index/% | Modulus of Fineness |
|-----------|---------------------------|------------------------|-------------------------|------------------|---------------------|
| Detection result | 3100 | 5.0 | 10.1 | 5.3 | 3.0 |

2.1.5. Rubber Particles (RP)

Scrap tire broken rubber particles were produced by Dujiangyan Huayi Rubber company (Chendu, China). The particle size of RP is 1~3 mm. The apparent density of the RP is 1120 kg/m$^3$. This paper used PDA for surface modification of RP. The RP surface was soaked in NaOH for 30 min to obtain an alkaline environment, in which potassium

permanganate reacted with dopamine to produce polydopamine (PDA)-coated $MnO_2$ nanotubes. PDA has strong adhesion properties. The modification methods of rubber particles are shown in Table 5.

**Table 5.** Modification methods of rubber particles.

| Modified Mode | Critical Step |
|---|---|
| PDA | Soak in 5% NaOH for 30 min; soak in 5% potassium permanganate solution for 30 min; soak in 2 mg/mL dopamine solution for 30 min |

### 2.1.6. Copper-Plated Steel Fiber (CPSF)

Its diameter is 0.18 mm. The length of the CPSF is 13 mm. Its tensile strength is 1800 MPa.

### 2.1.7. Expansion Agent

In this paper, MgO-type and II-Mg-type expansion agents were used. The MgO-type expansion agent was prepared by Wuhan Sanyuan Special Building Materials Co., Ltd. in Wuhan (China). The II-Mg-type expansion agent was mixed from II-type $Ca_4Al_6SO_{16}$-CaO expansion agent and MgO-type expansion agent with the proportion of 55%:45%. The main performance indexes of the expansion agent are shown in Table 6.

**Table 6.** Main performance indexes of expansion agent.

| Material | Specific Surface Area/(cm²/g) | Initial Setting Time/min | 7d Limit Expansion Rate in Water/% | 28d Compressive Strength/MPa |
|---|---|---|---|---|
| II-Mg Type | 3190 | 195 | 0.070 | 46.2 |

### 2.1.8. Water-Reducing Agent

Super-high-performance concrete special polycarboxylic acid high-efficiency water-reducing agent was used. The solid content of the water reducing agent was 35%. The water reduction rate of the water reducing agent was 35%.

### 2.2. Experimental Method Design

According to the benchmark mix ratio in Table 7, we prepared the net slurry of cementing material. The prepared slurry was placed in a 40 × 40 × 40 mm mold for molding (a total of 4 pieces). Then the mold was removed after maintenance for 1 day. After releasing the mold, the test blocks were subjected to standard curing, steam curing, dry curing and pressure steam curing, respectively. The conditions of standard curing: 20 °C ± 1 °C, relative humidity greater than 90%, curing to the age of 28 d. The conditions of steam curing: heating rate 15 °C/h, constant temperature 90 °C, cooling rate 10 °C/h, total curing time 72 h. The conditions of dry curing: heating rate 15 °C/h, constant temperature 90 °C, cooling rate 10 °C/h, sealing, total curing time 72 h. The conditions of pressure steam curing: 210 °C, 2 MPa, pressure steam curing time 8 h. The experimental organization design is shown in Figure 2.

**Table 7.** Initial mix proportion.

| Water-Binder Ratio | Raw Material/(kg/m³) | | | | Water Reducing Agent | Rubber Particle /vol% | Fibre /vol% | Expansion Agent /(kg/m³) |
|---|---|---|---|---|---|---|---|---|
| | Cement | Silica Fume | Microbead | High Titanium Heavy Slag Sand | | | | |
| 0.17 | 800 | 200 | 150 | 1150 | 3.1% | 10 | 2.6 | 70 |

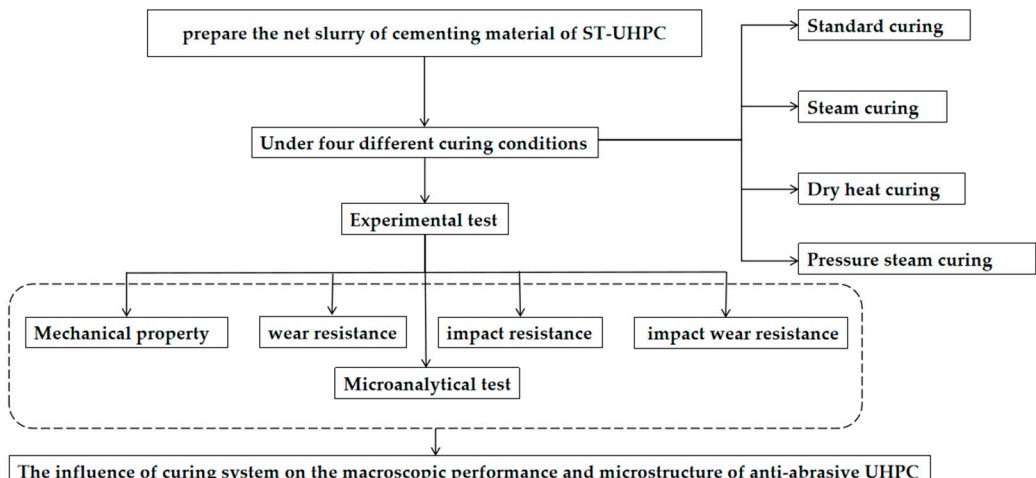

**Figure 2.** The experimental setup.

The test block was cured to the appropriate age. Then the test block was crushed to a particle size of less than 3 mm. After termination of hydration, the block was ground fine, screened by 0.075 mm, and dried for use. The samples under different curing systems were analyzed by XRD, SEM-EDS, 29Si NMR and 27Al NMR.

Anti-impact-wear UHPC was prepared according to Table 7. After mold release (1 d), anti-abrasive UHPC was maintained in standard curing, steam curing, dry heat curing and pressure steaming curing, respectively. The test blocks were cured to age (standard curing age is 28 days, steam curing and dry heat curing age is 4 days and pressure steam curing age is 2 days from forming). Then, the various performances of these test blocks were tested. The durability of the specimens was tested according to the requirements of the "Standard of Test Method for Long-Term Performance and Durability of Ordinary Concrete" (GBT50082-2009 [39]).

### 2.3. Experimental Method

(1)   Mechanical property test of UHPC:

The mechanical properties of concrete were tested in accordance with Active Powder Concrete (GB/T31387-2015 [40]) and Fiber Concrete Test Method (CECS13-2009 [41]).

(2)   Test and evaluation methods for wear resistance, impact resistance and impact wear resistance of UHPC:

The wear resistance of UHPC was measured by the underwater steel ball method. The impact resistance of UHPC was measured by the drop hammer impact method. These two methods evaluate the impact wear resistance of UHPC synthetically. The specific methods of the underwater steel ball method are as follows:

In accordance with relevant requirements of the "underwater steel ball method" in the Hydraulic Concrete Test Regulations (SL352-2006 [42]), UHPC was tested for impact-wear strength. The testing instrument was the SJA-1 concrete impact-wear (underwater steel

ball method) tester of Cangzhou Zhongjian precision Instrument Co., Ltd. in Cangzhou, China [21].

(3)　Microanalytical test method:

(1)　29Si NMR, 27Al NMR, XRD, SEM-EDS and chemically bound water were used to characterize the hydration products and microstructure of UHPC cementates at different curing ages.

(2)　The internal pore structure of UHPC cementate was studied with X-ray CT.

## 3. Results and Discussion

### 3.1. Effect of Curing System on Macroscopic Properties of UHPC

3.1.1. Mechanical Properties and Impact Resistance

Table 8 shows the mechanics, wear resistance and impact resistance of UHPC under different curing systems. Under standard curing, the compressive strength, flexural strength, impact strength and final crack impact energy of UHPC materials are 144.5 MPa, 26.0 MPa, 167 h/(kg/m$^2$) and 162 kJ, respectively. Compared with the standard curing conditions, the compressive strength, flexural tensile strength, impact wear strength and final crack impact energy of UHPC increased, respectively, by 2.62%, 1.92%, 1.20% and 1.85% under 90 °C steam curing. Compared with the standard curing conditions, the performance of UHPC increased by 14.60%, 11.92%, 2.40% and 3.80%, respectively, under the pressure steam curing at 210 °C and 2 MPa. The reasons for the difference in compressive and flexural strength and impact strength are as follows.

**Table 8.** Effect of curing system on UHPC performance.

| Group | Curing System | Compressive Strength/MPa | Flexural Tensile Strength/MPa | Abrasion Resistance Strength/h × (kg/m$^2$)$^{-1}$ | Impact Work of Terminal Crack/kJ |
|---|---|---|---|---|---|
| Y1 | Standard curing | 144.5 | 26.0 | 167 | 162 |
| Y2 | Steam curing | 148.3 | 26.5 | 169 | 165 |
| Y3 | Dry heat curing | 126.7 | 23.4 | 152 | 145 |
| Y4 | Pressure steam curing | 165.6 | 29.1 | 171 | 168 |

Compared with standard curing, steam curing can promote hydration of the cementing material. Then, promoting the hydration of the cementing material could improve the mechanical properties of UHPC. But compared with standard curing, the mechanical properties, wear resistance and impact resistance of UHPC show little difference under steam curing at 90 °C. The reason is that steam curing causes the rapid release of water in the internal curing aggregate of UHPC. This results in the increase of the interface between the internal curing aggregate of UHPC and UHPC slurry, and results in the increase of harmful pores in the UHPC matrix, as shown in Figure 3. Compared with the standard curing condition, the porosity of the UHPC matrix increased by about 26% and the number of harmful pores increased obviously under steam curing. Finally, the positive effect of steam curing on the mechanical properties of UHPC was offset.

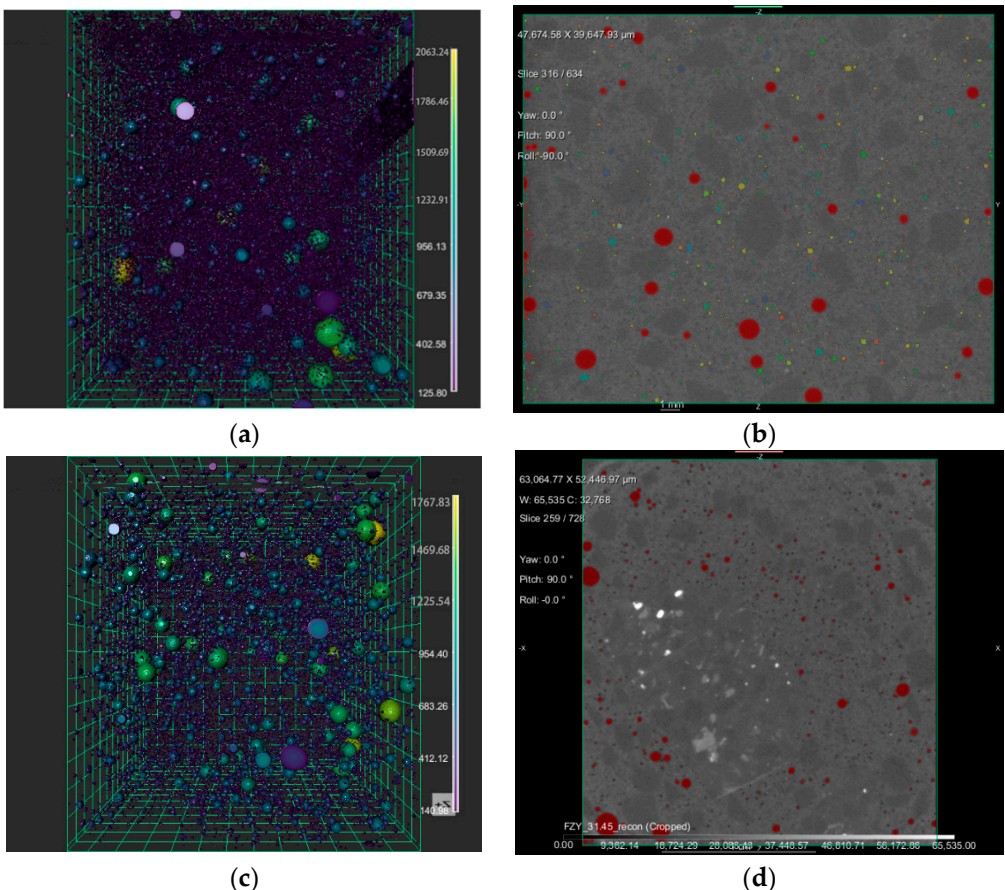

**Figure 3.** X-CT three-dimensional and two-dimensional sectional hole structure visualization maps of UHPC under steam curing and standard curing system. (**a**) X-CT three-dimensional pore structure under steam curing condition (the porosity is about 2.9%); (**b**) X-ray CT two-dimensional structure of broken face under steam curing condition; (**c**) X-CT three-dimensional pore structure under standard curing conditions (about 2.3% porosity); (**d**) X-ray CT two-dimensional structure of severed face under standard curing conditions.

Compared with steam curing, pressure steam curing at 210 °C and 2 MPa can significantly improve the mechanical properties, wear resistance and impact resistance of UHPC, although the interface between internal curing aggregate and UHPC slurry and harmful pores in UHPC matrix increase under pressure steam curing. From another perspective, pressure steam curing can significantly improve the hydration degree of UHPC cementing material. Under pressure steam curing, MCL and Al[4]/Si of C-A-S-H gel were increased, while Ca/Si were decreased. At the same time, the conversion of C-A-S-H gelling to tobermorite was promoted.

Compared with standard curing, the mechanical properties, wear resistance and impact resistance of UHPC were decreased significantly under dry heat curing. The reason is that there is non-exchange of water between the UHPC matrix and external water under dry heat curing. The non-exchange of water accelerated the water evaporation of UHPC matrix, therefore reducing the hydration degree of UHPC matrix and finally increasing the porosity and the number of harmful holes of UHPC slurry.

### 3.1.2. Durability

According to the Test Method Standard for Long-Term Performance and Durability of Ordinary Concrete (GB/T 50082-2009 [39]), the fast chloride ion mobility coefficient method (RCM method) and electric flux method were used to test the chloride ion penetration resistance of UHPC under different curing systems. The specimen used for testing was the

cylindrical mortar specimen with a diameter of 100 mm and a height of 50 mm. Because the electrical conductivity of the steel fibers would affect the test results, no fiber was added in the test. The test results are shown in the table below.

As can be seen from Table 9, the diffusion coefficient/electric flux of chloride ion is as follows: dry heat curing at 90 °C > standard curing at 20 °C > steam curing at 90 °C > pressure steam curing at 210 °C and 2 MPa. Under the four curing systems, the RCM rating of UHPC is all RCm-V, and the electric flux is Qs-V, with a small overall gap. All of the UHPCs have excellent chloride ion erosion resistance under the four curing systems. Under the four curing systems, all of the UHPCs are better than C60 anti-impact concrete prepared with high titanium slag sand.

**Table 9.** Impermeability test results.

| Group | Anti-Ionic Diffusion Coefficient/ ($\times 10$–12 m$^2$/s) | RCM Rating | Electric Flux/(C) | Rating of Electric Flux | Remarks |
|---|---|---|---|---|---|
| Y1 | 0.03 | RCM-V | 75 | Qs-V | Standard curing |
| Y2 | 0.02 | RCM-V | 70 | Qs-V | Steam curing |
| Y3 | 0.05 | RCM-V | 85 | Qs-V | Dry heat curing |
| Y4 | 0.01 | RCM-V | 60 | Qs-V | Pressure steam curing |
| Y5 [30] | 1.71 | RCM-IV | 715 | Qs-IV | / |

The wet-dry cycle method was used to test the resistance to sulfate ion penetration of UHPC with high impact resistance. The test results are shown in Table 10. The UHPC materials of Y1~Y4 group underwent 150 wet and dry cycles (Y5 group had 90 wet and dry cycles). The sulfate resistance levels of ST-UHPC reached KS150. The levels of KS150 indicated that curing system had little effect on the sulfate resistance of ST-UHPC. Because the water–binder ratio of UHPC is very low, the structure is dense, and there is no coarse aggregate to generate a large interfacial transition zone in UHPC. Therefore, it is difficult for sulfate ions to penetrate into the interior of UHPC.

**Table 10.** Test results of sulfate resistance.

| Group | Number of Wet and Dry Cycles | Corrosion Resistance Factor | Grade of Assessment | Remarks |
|---|---|---|---|---|
| Y1 | 150 | 1.02 | KS150 | Standard curing |
| Y2 | 150 | 1.03 | KS150 | Steam curing |
| Y3 | 150 | 0.99 | KS150 | Dry heat curing |
| Y4 | 150 | 1.05 | KS150 | Pressure steam curing |
| Y5 [30] | 90 | 0.83 | KS90 | / |

The freeze-thaw method was used to test the freeze-resistance of UHPC under different curing systems. The test results are shown in Table 11. The advantages and disadvantages of frost resistance are as follows: 90 °C dry heat curing < 20 °C standard curing < 90 °C steam curing < 210 °C 2 MPa pressure steam curing. In general, Y1~Y4 has good freezing–thawing resistance; the mass loss of 300 freezing–thawing cycles of Y1~Y4 is less than 0.20% (group Y5:1.3%). The relative dynamic elastic modulus of Y1~Y4 is greater than 95% (group Y5:82.1%). The main reason is that the water–binder ratio of UHPC is low, and the content of free water inside is very low. Part of the reason for the freeze-thaw cycle failure is caused

by the water inside the concrete causing it to freeze and expand. Therefore, UHPC with a low water–binder ratio can more easily resist freeze-thaw damage.

**Table 11.** Results of UHPC frost resistance test.

| Group | Test Item | Number of Freeze-Thaw Cycles/Times | | | | |
|---|---|---|---|---|---|---|
| | | 0 | 50 | 100 | 200 | 300 |
| Y1 | Mass loss/% | 0 | 0.08 | 0.11 | 0.14 | 0.18 |
| | Relative dynamic elastic modulus/% | 100 | 99.8 | 99.6 | 99.3 | 99.0 |
| Y2 | Mass loss/% | 0 | 0.07 | 0.11 | 0.13 | 0.17 |
| | Relative dynamic elastic modulus/% | 100 | 99.9 | 99.7 | 99.4 | 99.1 |
| Y3 | Mass loss/% | 0 | 0.09 | 0.13 | 0.17 | 0.20 |
| | Relative dynamic elastic modulus/% | 100 | 99.4 | 98.3 | 97.4 | 96.1 |
| Y4 | Mass loss/% | 0 | 0.06 | 0.10 | 0.12 | 0.15 |
| | Relative dynamic elastic modulus/% | 100 | 99.9 | 99.8 | 99.5 | 99.3 |
| Y5 [43] | Mass loss/% | 0 | 0.1 | 0.3 | 0.9 | 1.3 |
| | Relative dynamic elastic modulus/% | 100 | 99.1 | 96.3 | 86.4 | 82.1 |

*3.2. Effect of Curing System on Microstructure of UHPC*

3.2.1. Effect of Hydration Product Composition of Cementing Slurry

Figure 4 shows 28d XRD patterns of UHPC under four curing systems: standard curing, 90 °C steam curing, 90 °C dry heat curing and 210 °C 2 MPa pressure steam curing.

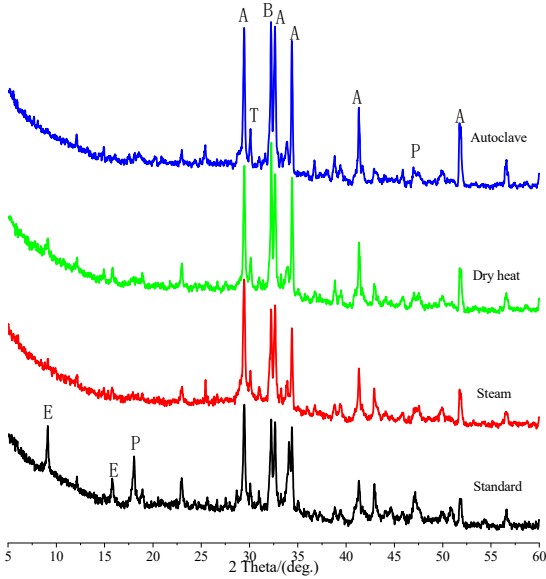

**Figure 4.** 28d XRD patterns of UHPC under different curing systems (A:$C_3S$, B:$C_2S$, P: Ca $(OH)_2$, E: AFt, T: tobermorite).

As shown in Figure 4, the phase composition of 28 d UHPC cementing slurry included $C_3S$, $C_2S$, $Ca(OH)_2$ and C-A-S-H gels under four curing systems. C-A-S-H gels were Amorphous and not shown on the XRD pattern. The peak shape of the unhydrated minerals ($C_3S$ and $C_2S$) is sharp and prominent, indicating the low hydration degree of

UHPC. The reason for the low hydration degree is the low water–binder ratio of UHPC (0.17) and the insufficient reaction of the cementing materials. In addition, after steam-curing at 2 MPa at 210 °C, an obvious tobermorite peak appears in the XRD pattern (as shown in Figure 5). The tobermorite peak indicates that pressure steam curing can promote the transformation of C-A-S-H gel into tobermorite. The above phenomenon is consistent with existing research results [44,45].

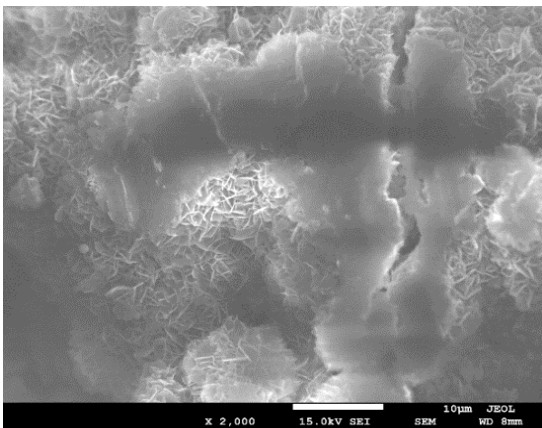

**Figure 5.** SEM images of cementitious paste under autoclave curing.

When comparing the XRD patterns of UHPC under the standard curing and the other three curing systems, the $Ca(OH)_2$ diffraction peak is more prominent under standard curing. This shows that high temperature and high pressure promote the hydration of cement and the pozzolanic effect of mineral admixtures. Under standard curing, the most prominent position of the $Ca(OH)_2$ diffraction peak is 2θ (18.089°). At 2θ (9.091°) and 15.784°, AFt diffraction peaks existed in XRD patterns under standard curing, while no obvious diffraction peaks existed in other curing systems. This is because AFt tends to decompose at high temperatures [46,47], so maintenance systems of high-temperature conditions cannot easily generate AFt. In addition, AFt crystallization requires a large amount of free water, while UHPC has a very low water–binder ratio, so it is not easy for UHPC to generate AFt [46,47].

3.2.2. Effect of Hydration Degree of Cementitious Slurry

The hydration product composition of UHPC slurry is quite different under different curing systems. Therefore, the curing system will have a certain impact on the hydration process of slurry. Therefore, the $^{29}$Si NMR test result was used to study the hydration degree of UHPC under different curing systems, and the influence of curing systems on the hydration process of slurry was further studied.

Figure 6 shows the $^{29}$Si NMR spectra of UHPC under different curing systems. The measured curves, fitting curves and curves of different peaking results of the spectra, respectively, are given in the different figures. Table 12 shows the results of the deconvolution calculation in Figure 6. Figure 7 shows the degree of hydration under different curing systems.

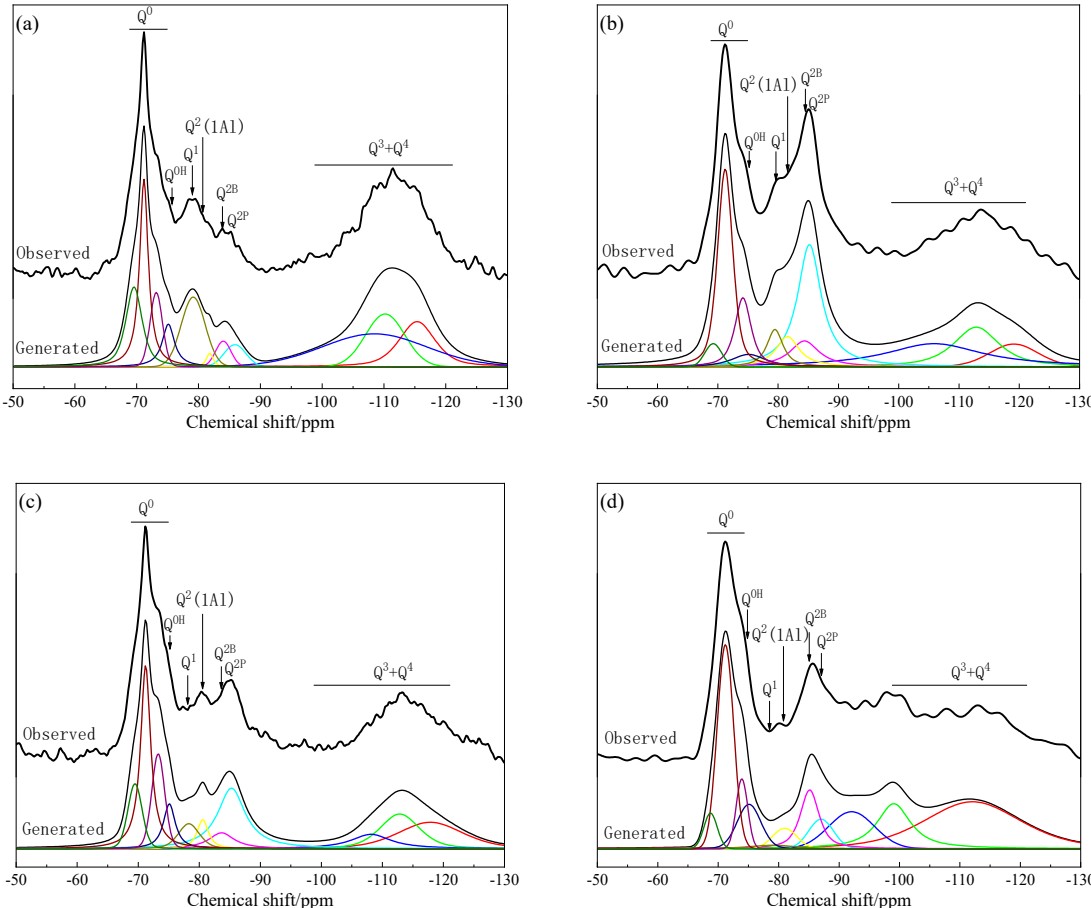

**Figure 6.** $^{29}$Si NMR spectra of UHPC under (**a**) standard curing system, (**b**) steam curing system, (**c**) dry heat curing system and (**d**) autoclave curing system.

**Table 12.** Deconvolution results of $^{29}$Si NMR spectra of UHPC under different curing systems.

| Group | $Q^n$ Relative Strength Value I/% | | | | | | | $\alpha_C$/% | MCL | Al[4]/Si |
|---|---|---|---|---|---|---|---|---|---|---|
| | $Q^0$ | $Q^{0H}$ | $Q^1$ | $Q^2$(1Al) | $Q^{2B}$ | $Q^{2P}$ | $Q^3 + Q^4$ | | | |
| Standard | 33.34 | 4.85 | 5.52 | 2.40 | 2.51 | 3.04 | 48.34 | 30.92 | 5.32 | 0.089 |
| steaming | 28.23 | 2.88 | 12.31 | 9.93 | 5.61 | 21.68 | 19.36 | 41.50 | 8.85 | 0.100 |
| Dry heat | 35.55 | 5.41 | 11.37 | 7.17 | 4.20 | 17.17 | 19.13 | 26.34 | 7.65 | 0.090 |
| Pressure steaming | 27.01 | 6.38 | 4.86 | 6.11 | 8.36 | 4.27 | 43.01 | 44.03 | 10.97 | 0.129 |

As shown in Figure 7, when the raw materials are the same, the hydration degree of cement in UHPC is as follows: 90 °C dry heat curing < standard curing < 90 °C steam curing < 210 °C 2 MPa pressure steam curing. Compared with standard curing, the cement hydration degree of UHPC increased by 34.22% and 42.38%, respectively, under 90 °C steam curing and 210 °C 2 MPa pressure steam curing. Therefore, 90 °C steam curing and 210 °C 2 MPa pressure steam curing significantly promoted the cement hydration process. Compared with the standard curing at 20 °C, the cement hydration degree of UHPC decreased by 14.81% under 90 °C dry curing. Therefore, dry curing at 90 °C is not conducive to the hydration of UHPC, because dry curing provides no water supply for later curing, and the water–binder ratio of UHPC is very low.

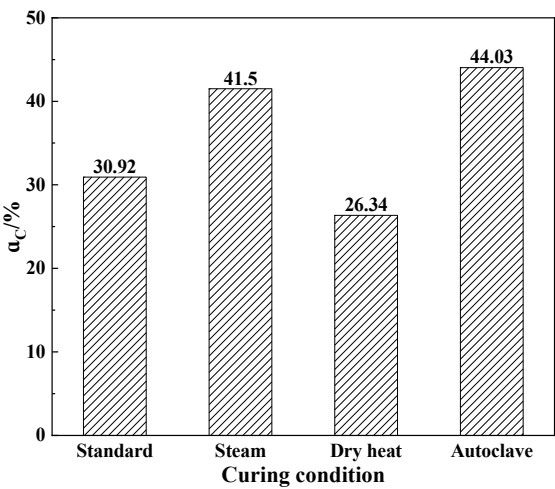

**Figure 7.** Hydration degree of UHPC under different curing systems.

3.2.3. Effect on the Microstructure of Cementitious Slurry

(1) C-A-S-H gel MCL and Al[4]/Si

Figure 8 shows the MCL and Al[4]/Si of UHPC under different curing systems. According to Figure 8a, the average molecular chain length of the C-A-S-H gel of the UHPC material is as follows: standard curing < 90 °C dry curing < 90 °C steam curing < 210 °C 2 MPa pressure steam curing. The results indicated that high temperature and high pressure could improve the degree of polymerization of the C-A-S-H gel's silica–oxygen chain. Heat curing will lead to the transformation of $Q^1$ to $Q^2$ on the silica–oxygen chain of the C-A-S-H gel, eventually increasing the average molecular chain length of the gel's silica–oxygen chain. The above phenomenon is consistent with the research results of Hu Chenguang [48].

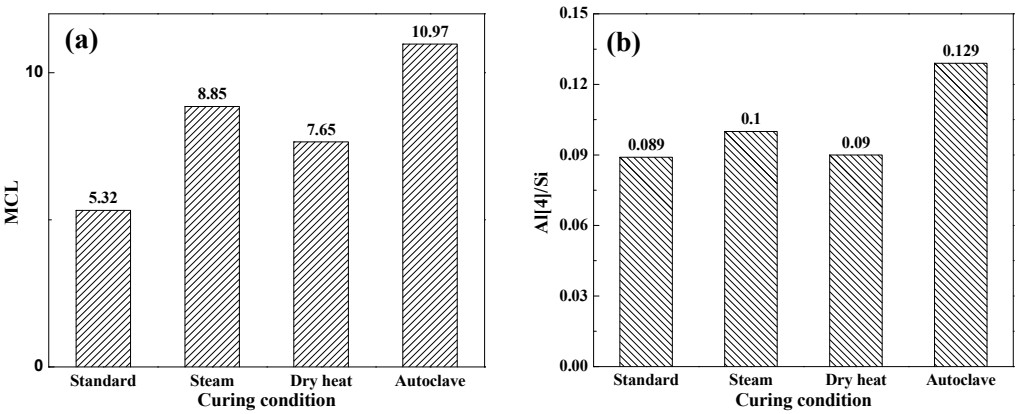

**Figure 8.** (**a**) MCL and (**b**) Al[4]/Si of UHPC under different curing systems.

As can be seen from Figure 8b, the Al[4]/Si of the C-A-S-H gel of the UHPC material was as follows: standard curing < 90 °C dry thermal curing < 90 °C steam curing < 210 °C 2 MPa pressure steam curing. As shown in Figure 6b, Al[4]/Si are 0.089, 0.100, 0.090 and 0.129, respectively, under standard curing, 90 °C steam curing, 90 °C dry heat curing and 210 °C 2 MPa pressure steam curing. Therefore, high temperature (high pressure) will promote the substitution of $[AlO_4]$ by $[SiO_4]$. In conclusion, hot curing and especially pressure steam curing can enhance the Al doping effect of UHPC slurry.

(2) Ca/Si

Figure 9 shows the SEM-EDS spectra of slurry under different curing systems. As shown in Figure 9, the Ca/Si of UHPC C-A-S-H gel is 2.15 under standard curing. The

Ca/Si of C-A-S-H gel (or tobermorite) is 1.37, 1.75 and 1.01, respectively, under steam curing at 90 °C, dry heat curing at 90 °C and pressure steam curing at 210 °C 2 MPa. Compared with standard curing, the Ca/Si of C-A-S-H gel (or tobermorite) decreased by 36.28%, 18.60% and 53.02% under the other three curing systems, respectively. The EDS results show that high temperature and high pressure can cause Ca loss in the main layer of the silicate structure and Si increase in the silica–oxygen chain. Ca loss and Si increase promotes the silica–oxygen chain connection, thus increasing the molecular chain length.

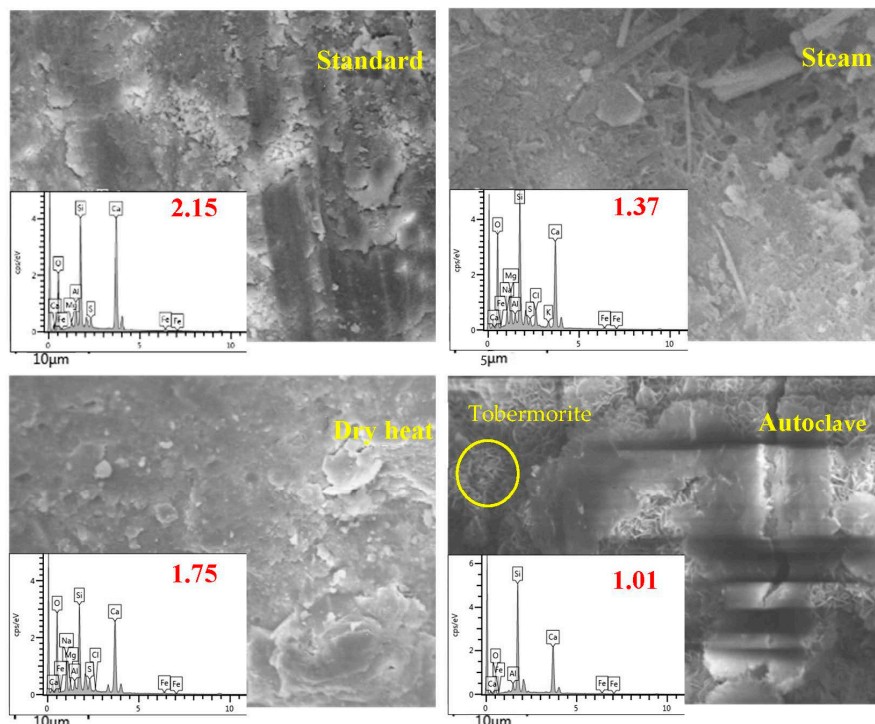

**Figure 9.** SEM-EDS spectrum of cementitious paste under different curing systems.

3.2.4. Effect of $Al^{3+}$ Substitution of $Si^{4+}$ in Cementitious Slurry

Figure 10 shows the 27Al NMR spectra of UHPC under different curing systems. Table 13 shows the results of the deconvolution calculation in Figure 10. Al[4]-C was formed by $[AlO_4]$ replacing $[SiO_4]$ in the C-A-S-H gel. As shown in Figure 10, Al[4]-C was 4.45, 7.04, 6.23 and 14.84, respectively, under standard curing, 90 °C steam curing, 90 °C dry heat curing and 210 °C 2 MPa pressure steam curing. These data indicate that high temperature can promote Al doping in UHPC slurry, and that pressure steam curing at 210 °C and 2 MPa has the best effect on Al doping. $Al^{3+}$ generates C-A-H, and then AFt is generated with $SO_4^{2-}$ in the matrix. Thanks to the active role of AFt in promoting the early strength development of cement and compensating for shrinkage and the Al doping in C-S-A-H, $Al^{3+}$ replaces $Si^{4+}$ to greatly improve the performance of UHPC.

**Table 13.** Deconvolution results of $^{27}$Al NMR spectra of UHPC under different curing systems.

| Curing Mode | Al[4]-C | Al[4]-Others | Al[5] | Al[6]-E | Al[6]-M | Al[6]-T |
|---|---|---|---|---|---|---|
| Standard | 4.45 | 22.17 | 9.17 | 8.26 | 41.52 | 14.43 |
| steaming | 7.04 | 40.36 | 9.39 | 15.99 | 14.49 | 12.73 |
| Dry heat | 6.23 | 58.66 | 9.28 | 2.61 | 16.84 | 6.38 |
| Pressure steaming | 14.84 | 30.62 | 6.31 | 5.65 | 18.66 | 23.92 |

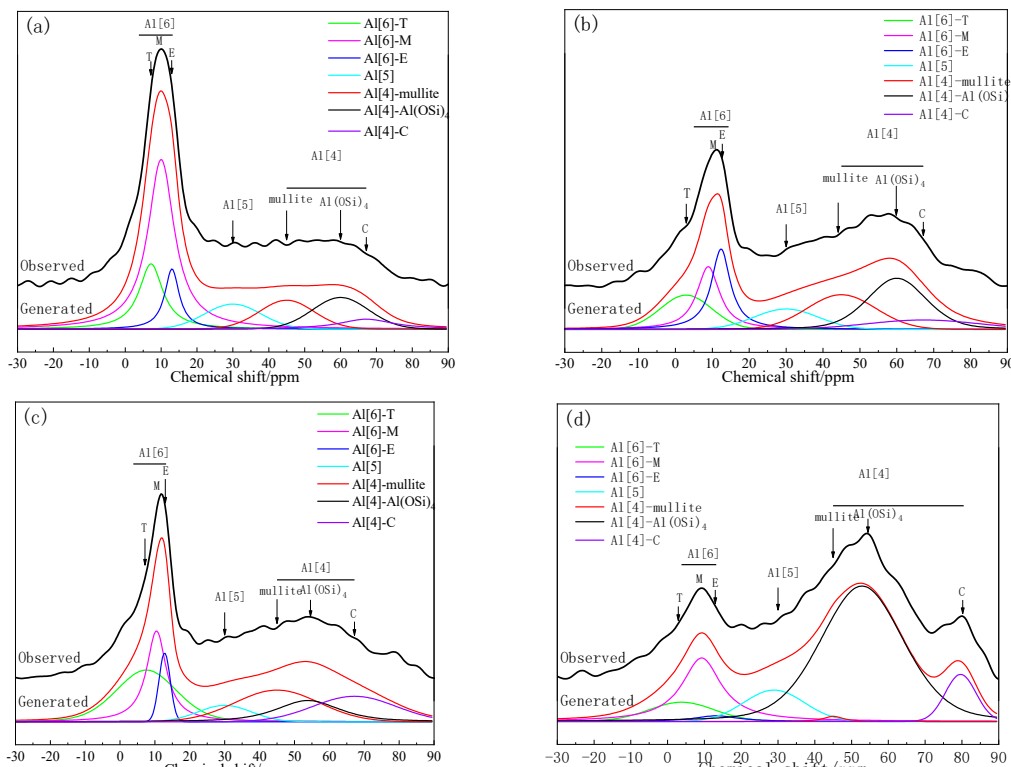

**Figure 10.** $^{27}$Al NMR spectra of UHPC under (**a**) standard curing system, (**b**) steam curing system, (**c**) autoclave curing system and (**d**) dry heat curing system.

### 3.3. Analysis on the Influence Mechanism of Curing System on Abrasion Resistance UHPC Properties

The influence mechanism of different curing systems on the properties of anti-abrasive UHPC is shown in Figure 11.

The main raw materials of anti-abrasive UHPC include cement, fly ash beads, silica fume, expansion agent, high titanium heavy slag sand, modified rubber particles, copper-plated steel fiber and admixture, etc. before hydration reaction. Different from the ordinary UHPC, the anti-abrasive UHPC uses the high titanium heavy slag sand. The high titanium heavy slag sand is prewet with water and high-strength and porous wear resistance. Different curing systems have an effect on the hydration products of UHPC. Compared with standard curing, high temperature will promote AFt decomposition, and pressure steam curing will promote the transformation of C-A-S-H gel into tobermorite. In addition, high temperature will evaporate the prewet water quickly in the saturated prewet porous high titanium heavy slag sand. It will increase the interface between the high titanium heavy slag sand and the UHPC slurry. In addition, reduction of the prewet water will increase the porosity and harmful pores of UHPC matrix. The influence mechanism on macro- and micro-properties of UHPC under different curing systems is different, and the analysis is as follows:

(1) Compared with standard curing, due to no water exchange with the outside, dry curing at 90 °C accelerated the water evaporation of the UHPC matrix. The reduction of water decreased the hydration degree and MCL of the UHPC slurry. The reduction of hydration degree will impair the macro-performance of UHPC. Therefore, dry curing at 90 °C deteriorated the macro-performance of UHPC.

(2) Compared with standard curing, steam pressure curing significantly improved the hydration degree of UHPC cementing material, increased the MCL and Al[4]/Si of C-A-S-H gel and decreased Ca/Si. In addition, steam pressure curing formed more Al-doped C-A-S-H gel. The more obvious the aluminum doping effect, the better the mechanical properties and wear resistance of UHPC will be [49]. At the same

time, steam pressure curing promoted the conversion of C-A-S-H gel to tobermorite. Because tobermorite will promote the macro-performance of UHPC, steam pressure curing significantly improved the macro-performance of UHPC.

(3)　Compared with standard curing, steam curing improved the hydration degree of UHPC cementing material, increased the MCL and Al[4]/Si of C-A-S-H gel and decreased Ca/Si. However, pressure steam curing results in the rapid release of water in the curing aggregate in UHPC. The release of water increases the interface between the aggregate and the UHPC slurry. In addition, the release of water increases the porosity of the UHPC matrix by about 26% and causes an obvious increase in harmful pores. The positive effect of steam curing on improving mechanical properties of UHPC is offset. Therefore, there is little difference between the macro-performance of UHPC under 90 °C steam curing and standard curing conditions.

(4)　In summary, non-steaming and normal temperature moisturizing maintenance should be adopted for anti-abrasive UHPC.

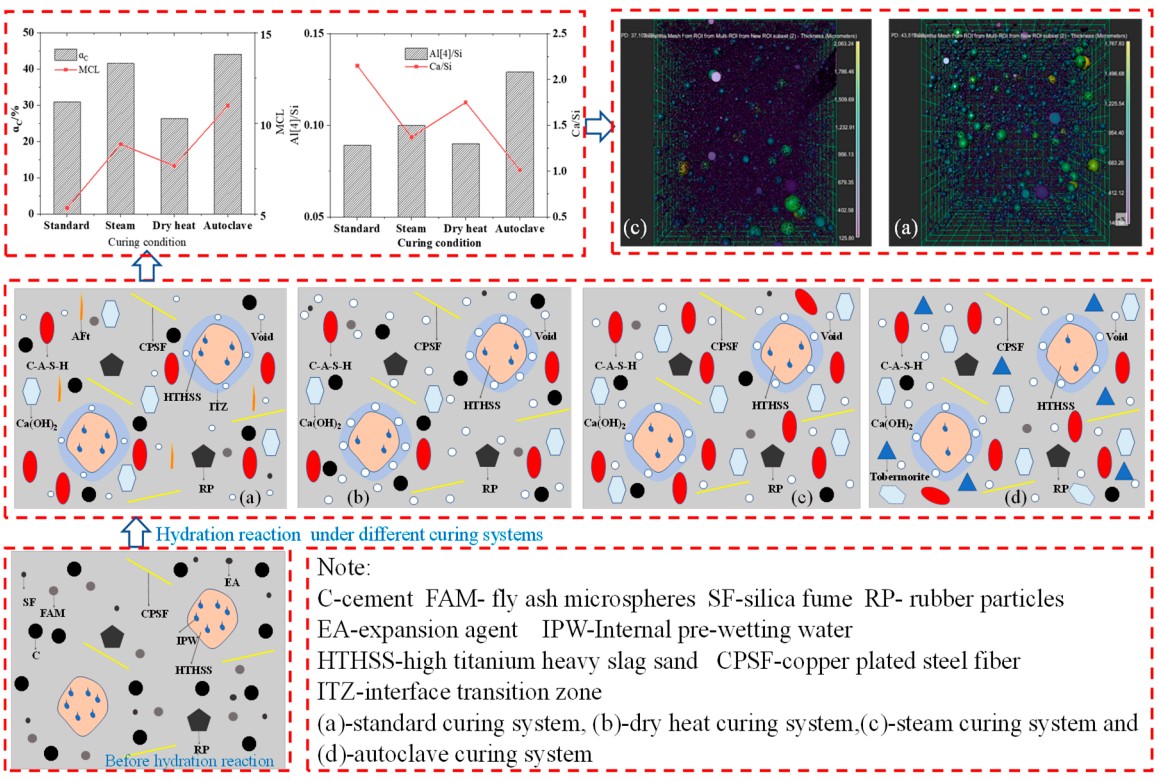

**Figure 11.** Schematic diagram of UHPC influence mechanism analysis under different curing systems.

## 4. Conclusions

(1)　Different curing systems have an impact on the hydration products of UHPC cementation. Pressure steam curing at 210 °C and 2 MPa will promote the transformation of C-A-S-H gel into tobermorite. High temperature will promote the decomposition of AFt.

(2)　Compared with standard curing, steam curing at 90 °C improved the hydration degree of cementing material in UHPC. And steam curing at 90 °C increased the MCL and Al[4]/Si of C-A-S-H gel. But steam curing at 90 °C would increase the interface between internal curing aggregate and UHPC cementing slurry. Steam curing at 90 °C would increase the harmful pores in the UHPC matrix. Compared with standard curing, steam curing at 90 °C showed little difference in the mechanical properties, wear resistance, impact resistance and durability of UHPC.

(3)　Compared with standard curing, pressure steam curing significantly improved the hydration degree of UHPC cementing material, increased the MCL and Al[4]/Si of

C-A-S-H gel, and decreased Ca/Si. At the same time, the conversion of C-A-S-H gelling to tobermorite was promoted. The mechanical properties, wear resistance, impact resistance and durability of UHPC can be improved by pressure steam curing at 210 °C and 2 MPa.

(4)  Based on the influence of different curing systems on the performance of UHPC, considering the construction and curing conditions of anti-abrasive protection of piers, it is suitable to adopt non-steaming and normal temperature moisturizing curing for anti-abrasive UHPC.

(5)  This study fills in the mechanism of the influence of various curing conditions on the microstructure of UHPC. Under different curing conditions, water evaporation in porous structures will affect the body strength. According to these two points, the prewetting of aggregate and curing systems can be adjusted, so that impact-wear UHPC can be applied in gradient.

**Author Contributions:** Methodology, data curation, writing—original draft, writing—review and editing, J.L.; funding acquisition, supervision, project administration, resources, Z.Y.; data curation, visualization, F.X.; project administration, Z.G.; software, Q.D. All authors have read and agreed to the published version of the manuscript.

**Funding:** This study was supported by the Open State Key Laboratory of New Textile Materials and Advanced Processing Technologies. Financial support from the Application Foundation Frontier Project of Wuhan Science and Technology Bureau (2022013988065200) was gratefully acknowledged. The authors are thankful for the support of the Basic and Applied Basic Research Foundation of Guangdong Province (2022A1515010508). This study was also supported by the Science and Technology Project of Hubei Transportation Department (2022-11-6-1) and the Science and Technology Project of Shandong Hi-speed Group Co., Ltd. (HSB2020119).

**Institutional Review Board Statement:** Not applicable.

**Informed Consent Statement:** Not applicable.

**Data Availability Statement:** Some or all data, models, or code that support the findings of this study are available from the corresponding author upon reasonable request.

**Conflicts of Interest:** The authors declare no conflict of interest.

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
