# Peer review of "The Influence of Curing System on the Macroscopic Performance and Microstructure of Anti-Abrasive UHPC"

_coatings, doi:10.3390/coatings14010045_

Round 1

Reviewer 1 Report

Comments and Suggestions for Authors

The paper investigates the efficacy of anti-abrasive Ultra-High-Performance Concrete (ST-UHPC) in mudslide-prone areas of western China. The authors obtained satisfactory results. The manuscript can be accepted after the following comments.

1.      The abstract is too long. Summarize the abstract by encompassing the study's importance, the research issue addressed, the employed methodology, key findings, and potential future implications of the results.

2.      The introduction of the current study lacks a concise statement regarding the research gap and objectives. It is essential to clearly articulate the research problem, identify the gap in existing literature, and highlight the novel contributions of the study.

3.      In the main text, divide and elaborate on the key findings from references (5-16). Additionally, expand the reference list by incorporating more pertinent studies from recent years.

4.      The fourth paragraph in the introduction lacks connections to references. Add relevant references to the paragraph.

5.      In experimental program section, add relevant pictures of the materials and experimental setup.

6.      How does the substitution of Al3+ for Si4+ affect the performance of UHPC? Explain in detail.

7.      Can you provide more details about the surface modification methods used for the rubber particles? How was the modification carried out?

8.      The authors need to highlight the significance of the study's contributions to the broader field and identify the novel or innovative aspects of the research.

Comments on the Quality of English Language

Moderate editing of English language required

Author Response

Manuscript ID: coatings-2770683R1

Title: The influence of curing system on the macroscopic performance and microstructure of anti-abrasive UHPC

General Reply

We would like to thank you and the reviewers for your great help and amendment suggestions for the improvement of our manuscript. According to your advice, we have carefully revised our manuscript and highlighted the changes in the uploaded revised manuscript. The main corrections in the paper and the point-by-point responds to the reviewer′s comments are as following.

Specific Reply

1# Reviewer

  1. The abstract is too long. Summarize the abstract by encompassing the study's importance, the research issue addressed, the employed methodology, key findings, and potential future implications of the results.

Reply and how the paper is modified: 

Thanks very much for your instructive advice. We are very grateful to the reviewer for the valuable comment on our manuscript. We have abbreviated the abstract section and highlighted the importance and results of the study.

  1. The introduction of the current study lacks a concise statement regarding the research gap and objectives. It is essential to clearly articulate the research problem, identify the gap in existing literature, and highlight the novel contributions of the study.

Reply and how the paper is modified: 

Thanks very much for your instructive advice. We are very grateful to the reviewer for the valuable comment on our manuscript. In the penultimate paragraph of the introduction, we highlight the new contributions of this study and the gaps in this field.

  1. In the main text, divide and elaborate on the key findings from references (5-16). Additionally, expand the reference list by incorporating more pertinent studies from recent years.

Reply and how the paper is modified: 

Thanks very much for your instructive advice. We are very grateful to the reviewer for the valuable comment on our manuscript. We have divided the literature [5-16] into three parts, the corresponding references are also added.An additional 10 papers were added.

  1. The fourth paragraph in the introduction lacks connections to references. Add relevant references to the paragraph.

Reply and how the paper is modified: 

Thanks very much for your instructive advice. We are very grateful to the reviewer for the valuable comment on our manuscript.Because there are no specific criteria for accelerated maintenance, we summarize the methods of previous literature and cite them in the text.

  1. In experimental program section, add relevant pictures of the materials and experimental setup.

Reply and how the paper is modified:

Thanks very much for your instructive advice. We are very grateful to the reviewer for the valuable comment on our manuscript.We added the appearance diagram of raw materials and the test flow chart, as follows.

  • Cement            (b) Fly ash microbeads          (c) Silica fume

  1. How does the substitution of Al3+for Si4+ affect the performance of UHPC? Explain in detail.

Reply and how the paper is modified:

Thanks very much for your instructive advice. We are very grateful to the reviewer for the valuable comment on our manuscript. We added a paragraph in 2.3.4. Al3+ generates C-A-H, and then AFt is generated with SO2- 4 in the matrix. Thanks to the active role of AFt in promoting the early strength development of cement and compensating for shrinkage and the Al doping in C-S-A-H, Al3+ replaces Si4+ to greatly improve the performance of UHPC.

  1. Can you provide more details about the surface modification methods used for the rubber particles? How was the modification carried out?

Reply and how the paper is modified:

Thanks very much for your instructive advice. We are very grateful to the reviewer for the valuable comment on our manuscript.We added additional instructions: The RP surface was soaked in NaOH for 30 minutes to obtain an alkaline environment, in which potassium permanganate reacted with dopamine to produce polydopamine(PDA) coated MnO2 nanotubes. PDA has strong adhesion properties.

  1. The authors need to highlight the significance of the study's contributions to the broader field and identify the novel or innovative aspects of the research.

Reply and how the paper is modified:

Thanks very much for your instructive advice. We are very grateful to the reviewer for the valuable comment on our manuscript.We have deleted the conclusion (3), added the contribution of this study to the field and pointed out its novelty.

Reviewer 2 Report

Comments and Suggestions for Authors

I congratulate the authors on a very interesting research work and I would like to emphasize that, in the reviewer's opinion, the article is well written, but the following questions arose while reading:

Comments to be clarified in the reviewer's opinion:

1. Please explain or complete the research in the scope of the given data in the table "Physical properties of the three cementitious materials" regarding HFC1 and FAM in the scope of the given data: initial setting time/min, 3d compressive strength/MPa over 28d compressive strength/MPa.

2. Please explain why the increases in compressive and flexural strength and impact strength given in point 3.1.1 differ so much

In terms of frost resistance, I cannot find results on the impact on compressive strength - please explain

3. Please correct the data in table 12 because it is illegible to the reviewer and future reader

4. The conclusion from paper no. 3 is obvious and can be omitted

5. Please answer the question whether steam hardening under pressure at 2100C is possible in prefabrication plants?

Author Response

Manuscript ID: coatings-2770683R1

Title: The influence of curing system on the macroscopic performance and microstructure of anti-abrasive UHPC

General Reply

We would like to thank you and the reviewers for your great help and amendment suggestions for the improvement of our manuscript. According to your advice, we have carefully revised our manuscript and highlighted the changes in the uploaded revised manuscript. The main corrections in the paper and the point-by-point responds to the reviewer′s comments are as following.

Specific Reply

2# Reviewer

  1. Please explain or complete the research in the scope of the given data in the table "Physical properties of the three cementitious materials" regarding HFC1 and FAM in the scope of the given data: initial setting time/min, 3d compressive strength/MPa over 28d compressive strength/MPa.

Reply and how the paper is modified:

Thanks very much for your instructive advice. We are very grateful to the reviewer for the valuable comment on our manuscript. Add 10% cement mass of FAM, SF into the cement, and then stir the sample for relevant performance testing.

  1. Please explain why the increases in compressive and flexural strength and impact strength given in point 3.1.1 differ so much.In terms of frost resistance, I cannot find results on the impact on compressive strength - please explain.

Reply and how the paper is modified:

Thanks very much for your instructive advice. We are very grateful to the reviewer for the valuable comment on our manuscript. In this paper, we give the reason for such a large difference in performance: the difference in pore structure and interface connection capacity caused by the maintenance system. We will continue to study the compressive strength of frost resistance.

  1. Please correct the data in table 12 because it is illegible to the reviewer and future reader

Reply and how the paper is modified:

Thanks very much for your instructive advice. We are very grateful to the reviewer for the valuable comment on our manuscript. This is our mistake, we have corrected the relevant data in the table 12, please check.

  1. The conclusion from paper no. 3 is obvious and can be omitted

Reply and how the paper is modified:

Thanks very much for your instructive advice. We are very grateful to the reviewer for the valuable comment on our manuscript. We have deleted the conclusion (3), added the contribution of this study to the field and pointed out its novelty.

  1. Please answer the question whether steam hardening under pressure at 210℃is possible in prefabrication plants?

Reply and how the paper is modified:

Thanks very much for your instructive advice. We are very grateful to the reviewer for the valuable comment on our manuscript. Because the test is carried out in strict accordance with the specified pressure and temperature, it can also be steam hardened in prefabrication plants.
